# Vaccination in Inflammatory Bowel Disease: Utility and Future Perspective

**Giovanni Casella** [1,*]**, Fabio Ingravalle** [2]**, Adriana Ingravalle** [3]**, Claudio Monti** [1]**, Fulvio Bonetti** [1]**, Federica De Salvatore** [4]**, Vincenzo Villanacci** [4] **and Aurelio Limonta** [1]

1   General Practitioner Limbiate (Monza Brianza), ATS Lecco- Brianza, 20812 Limbiate, Italy; C.Monti_2012@libero.it (C.M.); fulvio.bonetti@tin.it (F.B.); auli@live.it (A.L.)
2   School of Specialization in Hygiene and Preventive Medicine, University of Tor Vergata, 00133 Rome, Italy; fabio.ingravalle@gmail.com
3   Department of Obstetrics and Gynecology, Campus Bio-Medico University, 00128 Rome, Italy; nana.ingra@gmail.com
4   Institute of Pathology, Spedali Civili Brescia, 25123 Brescia, Italy; federica.desalvatore@libero.it (F.D.); villanac@alice.it (V.V.)
\*   Correspondence: Caselgio@tiscali.it; Tel.: +39-02-9963149 or +39-347-3840325

**Abstract:** Inflammatory bowel disease (IBD) is an immune-mediated disease, which often require lifetime treatment with immunomodulators and immunosuppressive drugs. Both IBD and its treatments are associated with an increased risk of infectious disease and mortality. Several of these diseases are vaccine preventable and could be avoided, reducing morbidity and mortality. However, vaccination rates among patients with IBD are lower than in the general population and both patients and doctors are not fully aware of the problem. Education campaigns and well planned vaccination schemes are necessary to improve vaccination coverage in patients with IBD. Immunomodulators and immunosuppressive drugs may reduce the seroprotection levels. For this reason, new vaccination schemes are being studied in patients with IBD. It is therefore important to understand which and when vaccines can be administrated based on immunocompetence or immunosuppression of patients. Usually, live-attenuated vaccines should be avoided in immunosuppressed patients, so assessing vaccination status and planning vaccination before immunosuppressive treatments are pivotal to reduce infection risk. The aim of this review is to increase the awareness of the problem and provide a quick reference for vaccination plan tailoring, especially for gastroenterologists and primary care physicians, who have the skills and knowledge to implement vaccination strategies.

**Keywords:** vaccinations; inflammatory bowel disease; prevention

## 1. Introduction

Patients with inflammatory bowel disease (IBD) have a higher risk of infectious disease, because of an impaired immune system function, both for native and adaptive immune response, and due to immunosuppressive treatments. However, several of these diseases can be prevented by vaccination or at least their severity can be reduced [1].

Several studies have demonstrated that both immunocompromised and immunocompetent patients with IBD have a higher risk of infective disease. A systematic review and network meta-analysis points out that patients with IBD on immunosuppressive treatment have an increased risk of opportunistic infection, but not the risk of serious infections [2]. However, the authors suggested that the long-term safety profiles of immunosuppressive treatment in patients with IBD must be monitored [2]. A cohort prospective study evaluated the long-term safety of infliximab and other

therapies in Crohn's disease [3]. The results showed that narcotic analgesics, prednisone and infliximab treatments were associated with serious infections, as was moderate-to-severe disease activity [3]. Similarly, a retrospective cohort study demonstrated that patients with IBD had an increased risk of influenza compared to the general population and were more likely to require hospitalization. Moreover, steroid medications were significantly associated with influenza risk [4].

For these reasons, the American Council Gastroenterology (ACG) and European Crohn's and Colitis Organization (ECCO) developed guidelines to help primary care physicians and gastroenterologists to not hesitate to vaccinate their patients against IBD [5,6].

Melmed et al. [7] discovered, before the publication of the ACG and ECCO guidelines, that immunization against vaccine-preventable disease was uncommon, because of the lack of awareness for the risk of the disease and the concern about the vaccination's side effects. However, vaccination rates in the IBD population are still low, even after the publication of the ACG and ECCO guidelines. Martinelli et al. [8] demonstrated a poor immunization status at diagnosis in children with IBD, which was not followed by a proper vaccination catch-up. Vaccination rates at diagnosis were unsatisfactory for measles, mumps, and rubella (89.3%), Haemophilus influenzae (81.9%), meningococcus C (23.5%), chickenpox (18.4%), pneumococcus (18.6%), papillomavirus (5.9%), and rotavirus (1.9%) [8]. Similarly, García-Serrano C et al. [1] found that vaccination compliance did not exceed 65% for any of the vaccines analysed in their study. The authors suggested a greater awareness among both patients and physicians is pivotal to increase vaccination rates and reduce the risk of vaccine-preventable infections [1].

Moreover, vaccination strategies and schedules are continually updated and improved. The aim of this review is to summarize and report the literature and vaccination recommendations for patients with IBD and provide a useful guide in managing vaccination schedules.

## 2. Vaccination Program

An ideal vaccination program should start at the same time as the diagnosis of IBD and before the administration of immunosuppressive therapies. The first visit should be considered "the ideal time" to obtain a vaccination history of the patients with IBD. Vaccination history is a pivotal point to tailor an individual vaccination plan for each patient. If the vaccination history cannot be retrieved or the vaccinal coverage cannot be confirmed, antibodies for vaccine-preventable disease should be assessed [5,6].

Anamnesis, vaccination history, the patient's physical status and therapy are some of the most important information that a gastroenterologist or a primary care physician must have for tailoring a vaccination plan. As inactivated vaccines are safer than live-attenuated vaccines, they can be administrated even during immunosuppressive therapies. However, the most probable outcome will be an inadequate response to vaccination, with a low or absent level of antibodies. Instead, live-attenuated vaccines should not be administered during immunosuppressive treatments and their use must be evaluated in term of benefits and risks [5,6]. A vaccination schedule should be tailored at least on two different patient sub-groups: immunocompetent patients with IBD and immunocompromised patients with IBD (Table 1).

**Table 1.** When may the patient with inflammatory bowel disease (IBD) be considered immunocompromised?

| |
|---|
| The evaluation of immunological status and vaccination in a patient with IBD is pivotal before starting any immunosuppressive treatment and to perform an adequate vaccination program. |
| According to the European Crohn's and Colitis Organization (ECCO) guidelines [6], a patient with IBD should be considered immunocompromised in the following conditions:<br>　　1A) Treatment with Prednisone 20 mg for a period more than 2 weeks;<br>　　1B) Treatment with immunosuppressive drugs as Azathioprine, 6-Mercaptopurine and Methotrexate (low-dose immunosuppression);<br>　　1C) Treatment with calcineurin inhibitors, cyclosporin or tacrolimus; |

1D) Treatment with biologic drugs (anti-TNF) as infliximab, adalimumab, certolizumab Pegol and others;

1E) Malnourished patient;

1F) Patients with acquired immunodeficiency syndrome (HIV-Positive), Iposplenism, Asplenism.

According with the Centres for Disease Control and Prevention (CDC) general recommendations for vaccination and immunoprophylaxis [9], a patient can be defined immunocompromised as follows:

2A) Low-dose corticosteroid (<20 mg/day of prednisone or equivalent, short or long term or alternating days);

2B) Glucocorticoid replacement therapy in adrenal insufficiency;

2C) Topical steroids or intra-articular, intra-bursa, or intra-tendon steroid injection;

2D) Low-dose methotrexate (<0.4 mg/kg/week or <20 mg/week);

2E) Low-dose azathioprine (<3 mg/kg/day);

2F) Low-dose 6-mercaptopurine (<1.5 mg/kg/day).

According to the Center Disease Control general recommendations for vaccination and immunoprophylaxis [9], live-attenuated vaccines can be administered after the cessation of immunosuppressive/immunomodulatory treatment according on the pharmacodynamic features of the drugs. Live-attenuated vaccines should be delayed for at least:

3A) Five half-lives after the administration of biological agents or disease-modifying drugs (3–12 months);

3B) Four weeks after high-dose corticosteroid therapy (≥20 mg/day prednisone or equivalent, for longer than 2 weeks);

3C) Four weeks after etanercept and 3 months after other TNF inhibitors (infliximab, adalimumab);

3D) Four to twelve weeks after the doses of ≥0.4 mg/kg/week or ≥20 mg/week of methotrexate (expert opinion is that there is no need to wait for herpes zoster or varicella zoster vaccine if the patient is on doses of methotrexate lower than 0.4 mg/kg/week or 20 mg/week);

3E) Six to twelve months after rituximab (if possible, live vaccines should be delayed until the B cell count returns to normal levels.);

3F) Two years after leflunomide.

According to the present guidelines, all patients with IBD should consider vaccination against the following pathogenetic agents: Streptococcus pneumoniae (Pneumococcus), Neisseria meningitidis (Meningococcus), flu virus, hepatitis B virus (HBV), measles virus, mumps virus, rubella virus, varicella zoster virus (VZV), human papilloma virus (HPV), Clostridium tetani, Corynebacterium diphtheriae, poliovirus, hepatitis A virus, Vibrio cholerae, Salmonella typhi and yellow fever virus [5,6].

## 3. Inactivated Vaccines

### 3.1. Pneumococcal Vaccination

Pneumococcus (*Streptococcus pneumoniae*) is among the most important causes of serious diseases such as pneumonia, meningitis and sepsis in the adult population, as pointed out by the Center for Disease Control and Prevention (CDC) of United States of America [10]. Major risk factors for pneumococcal infection can be immunodeficiency, chronic diseases, functional or anatomic asplenia, cerebral spinal fluid leaks and cochlear implants [10]. In addition, in the next few years, pneumococcal infection will become increasingly dangerous, due to greater resistance to antibiotics [11]. Therefore, preventive measures such as vaccination will play an increasingly important role in the future. Nowadays, several research teams have assessed that patients with IBD have an increased risk both of infection [12,13] and mortality [3]. Pneumococcal vaccination is recommended in all patients with IBD according to the AGC and ECCO guidelines and vaccination should be given before the administration of immunomodulator therapies [5,6].

Several vaccine formulations are available, but the two recommended for IBD are Pneumococcal Conjugate 13 Valent (PCV13) vaccine (Prevnar®) and Pneumococcal Polysaccharide 23 Valent

(PPVS23) vaccine (Pneumovax®). PCV13 is a 13 valent conjugate vaccine, in which pneumococcal capsular polysaccharides are conjugated to highly immunogenic Cross-Reactive Material 197 (CRM197), which triggers the adaptive immune system and the generation of Memory B cells. While PPVS23 contains purified capsular polysaccharides from 23 pneumococcal serotypes that can stimulate specific IgM production by B Cells [14], in patients with IBD, the use of immunosuppressive drugs as corticosteroid, azathioprine, methotrexate and biological therapy may also impair the immune response to pneumococcal vaccination [15]. In particular, corticosteroids [16,17], anti-TNF$\alpha$ [16] and combination therapy (infliximab and azathioprine) [16] were confirmed to be associated with an impaired response to vaccination.

The Advisory Committee on Immunization Practices (ACIP) recommends the following vaccination schedule: one first dose of PVC13, followed by a second PPSV23 after at least 8 weeks and a booster dose of PP23 5 years after the first dose. In patients who have already received at least one dose of PPSV23, a dose of PVC13 should be given after almost a year. A booster dose is recommended at 65 years [10]. The administration of a single PCV13 dose was proved to be immunogenic and well tolerated in patients with IBD, although long-term immunogenicity has not yet been proven [18]. The vaccination rate in patients with IBD are low, but it could be increased by stressing its importance. A simple educational form given to the patient in waiting rooms has resulted in an increase in the pneumococcal vaccination rate from 21% to 32% [19].

## 3.2. Meningococcal Vaccination

Meningococcus (*Neisseria meningitidis*) is among the most noted bacteria, which can determine meningitis, sepsis and pneumonia, also in immunocompetent young adults. Risk factors for meningococcal disease include household crowding, military recruits, patients living in college dormitories, patients travelling to endemic areas, patients with chronic illness, active and passive smoking, medical history of viral infection, asplenic or immunodeficiency [20]. There are no data to sustain that patients with IBD without these risk factors have an increased risk to develop the disease. However, patients with at least one or more of these risk factors have an increased risk of developing meningococcal disease. Moreover, anti-meningococcal vaccination is recommended by the ACG guidelines in accordance with routine vaccination recommendations [5].

Several types of anti-meningococcal vaccines are available on the market, the most used including the vaccines against serogroups A, C, W and Y (Men-ACWY) [20] and against serogroup B (Men-B) [21]. The Men-ACWY vaccine is available in two different forms: a Meningococcal Quadrivalent Polysaccharide Vaccine (MPSV4) and a Meningococcal Quadrivalent Conjugate Vaccine (MCV4). No difference regarding the specific risk of meningococcal disease has been demonstrated in the scientific literature, therefore the guidelines recommend to vaccinate patients with IBD without risk factors, such as the general population. Nowadays, there are no indications for anti-meningococcal vaccination in adults without a risk factor with the Men-ACWY vaccine, while all adults between 16–23 years should receive two doses of the Men-B vaccine at 0 and 6 months [20,21]. In the adult population at risk, the suggested schedule for Men-ACWY is to receive a single dose or a 2-dose primary series based on the indication for vaccination with MCV4, 2 months apart [20]. The adult population, and children at 10 years old or more, at an increased risk for meningococcal disease should also receive three doses of the Men-B vaccine at 0, 1–2 and 6 months [21]. According to the ACIP's Meningococcal Vaccine Work Group, it is not known how long the vaccine protection will last, but a vaccinated patient with a prolonged risk of meningococcal disease should be revaccinated at 5-year intervals [22].

Regardless of the lack of evidence of the increased risk for patients with IBD and the vaccine effectiveness with or without immunosuppressive therapy, meningococcal vaccines are inactivated and can be administered to all patients with IBD unrelatedly to immunosuppression [5]. However, the ACG guidelines recommend vaccinating patients with IBD before the initiation of immune suppression when possible [5]. No data are available regarding vaccination rate against meningococcal disease in the IBD population.

### 3.3. Influenza Vaccination

Influenza (or seasonal flu) is among the most common diseases, which can be preventable through vaccination. Most of the patients who contract influenza will recover without complication, nevertheless influenza can evolve into serious complications such as hospitalization and death [23]. The risk of influenza infection and influenza complication is higher in patients with IBD rather than in the general population [4].

The influenza vaccine is available in five formulations: (1) a trivalent inactivated vaccine (IIV3); (2) a quadrivalent inactivated vaccine (IIV4); (3) a quadrivalent attenuated vaccine (LAIV4); (4) a quadrivalent recombinant influenza vaccine (RIV4); and 5) a live intranasal vaccine. According to the ACIP guidelines, IIV4 or RIV4 are suitable for healthy persons, however each formulation has its own indications and contraindications [23]. The European Crohn's and Colitis Organization (ECCO) recommends using the IIV3 formulation in all patients with IBD with and without immunosuppressive therapy [6]. However, the LAIV4 should only be used for healthy persons aged 5–49 years, and is not recommended for patients on immunomodulators [23]. Additionally, the live intranasal vaccine is not recommended for patients with IBD. The influenza vaccination should be scheduled annually as in the general population [6].

Several studies have investigated the impact of immunomodulator drugs on seroconversion. In particular, anti-TNF$\alpha$ and combined therapies are associated with a suboptimal response to influenza vaccination in both adults and in children with IBD. The seroconversion rate of the influenza vaccine in patients with IBD was 45% to 80% in adults and 33% to 85% in children [24–26]. Influenza vaccination is recommended, although the seroconversion levels achieved are lower than those obtained in the general population [5,6].

New vaccination strategies for influenza are part of an ongoing investigation. A recent study investigated the immunogenicity of IIV4 formulation in patients with IBD, and the formulation showed a good seroprotection and seroconversion rates, however immunogenicity was lower in patients receiving infliximab therapy [27]. Moreover, a recent RCT has demonstrated that a higher dose of the influenza vaccine in patients with IBD on anti-TNF$\alpha$ therapy is associated with a significantly higher post-immunization level of antibodies compared to a standard dose of the same influenza vaccine [28].

Nonetheless, the influenza vaccination is recommended, as several studies have pointed out that the vaccination rate in the IBD population is low. In fact, Sitte J et al. [29] found a vaccination rate of 34.7% in outpatients with IBD and Jackson BD et al. [30] found out a vaccination rate of 16% in their tertiary centre. Moreover, the rate of influenza vaccination could be increased, stressing its importance, even by a simple educational form given to the patient in waiting rooms. This strategy rises vaccination coverage from 23% to 47% [19].

### 3.4. Hepatitis B Virus (HBV) Vaccination

The prevalence of HBV infection in patients with IBD seems to be the same as in the general population across different countries [31,32]. Patients with IBD have a higher risk of developing or undergoing a reactivation of HBV, because of the immunosuppressive/biological treatments [33,34]. Specifically, treatment with two or more immunosuppressants was an independent predictor of HBV reactivation [33] and prolonged immunosuppression (>3 months) was an independent predictor of liver dysfunction, even if the prescription of immunosuppressants (corticosteroids, azathioprine/6-mercaptopurine or infliximab) was significantly lower in HBsAg-positive than HBsAg-negative patients with IBD [34].

Nevertheless, HBV serological status should be tested in all patients as soon as possible after the diagnosis, if it is unknown. Moreover, viremia should be assessed in HBV-positive patients. In patients tested negative for HBsAg, the vaccination should be administered as per the guidelines [5,6]. According to the ACIP guidelines, the vaccination schedule for adolescents (aged 11–19) should be three doses of 5 mcg of vaccine (Recombivax® or equivalent) at 0, 1 and 6 month and for adults (> 19 years) it should be three doses of 40mcg of vaccine (Recombivax® or equivalent) at 0, 1 and 6 months [27]. After the complete schedule, the seroprotection rates are > 90% in healthy adults aged <

40 years, while the seroprotection rate falls to 75% in persons aged 60 years [34]. According to the present guidelines, patients with IBD should follow the same schedule as the general population [5,6]. However, seroprotection levels achieved by patients with IBD are lower than in healthy controls, both in adults [35] and in children [36]. In a recent study, the impact of immunosuppressive treatment on seroprotection levels (HBsAb ≥ 10 IU/L) was analysed. The results showed that groups under treatment with methotrexate (37.1%) or infliximab (35.5%) are associated with a lower seroprotection rate compared to the reference arm (67.1%) and the seroprotection rate higher than 100 IU/l was achieved even by a smaller percentage of patients [36]. Moreover, the general seroprotection level in this study was far below the seroprotection level achieved in healthy adults, provided by the CDC [35,37].

The seroprotection level should be assessed 1–2 months after the last anti-HBV vaccine dose [6]. Different strategies were proposed to improve inadequate seroprotection levels (HBsAb < 10 IU/L) such as a fourth booster dose or a faster vaccination protocol. An effective strategy was proposed by Schillie et al. [36], where patients were vaccinated with a faster and higher protocol than the standard one (Engerix B® double dose at 0, 1 and 2 month). The double dose was associated with a better seroprotection rate than the standard one (OR, 4; 95% CI, 2–8; $p < 0.001$) [36]. Moreover, Pratt PK Jr et al. [38] demonstrated that patients with a prior HBV vaccination failure were more likely to develop adequate seroprotective levels of HBsAb following three additional vaccine doses, rather than one or two alone (OR 1.77, $p = 0.01$; OR 1.9, $p = 0.03$, respectively).

Although, according to the international literature, patients with IBD vaccinated against HBV were ranging from 12% to 49% in Europe [33,39,40], and the vaccination rate was reported to be lower in the US [33]. Moreover, only a fraction of patients enrolled in these studies have received a prior adequate HBV screening [33,39–41].

### 3.5. Hepatitis A Virus (HAV) Vaccination

Hepatitis A is among the most common worldwide infections, which is transmitted by the faecal–oral route, and generally it does not become chronic and its incidence was declining due to childhood vaccination. However, an increase in HAV infection during the period 2016–2018 in the US was reported and ACIP recommends vaccination for adults who plan travel to HAV-endemic countries, persons who use drugs, persons with chronic liver disease, and recently, persons experiencing homelessness [42].

For this reason, actual guidelines recommend HAV vaccination in patients with IBD, especially if the patients are immunocompromised, have chronic liver disease or live in/travel to HAV-endemic countries [5,6]. HAV vaccination can be performed with the same schedule of healthy persons, due to the vaccine being inactivated, safe and well tolerated [5,6]. The vaccination schedule recommends two doses of single antigen formulation (Havirx®) at 0 and at 6–12 months after the first dose (or Vaqta® at 0 and 6–18 months) [42]. Combined vaccination against both HAV and HBV can be safely administrated [6] and the vaccination schedule recommends three doses at 0, 1 and 6 months (Twinrix®); alternatively, an accelerated vaccination schedule recommends three doses at 0, 7 and 21–30 days plus one booster dose at 12 months (Twinrix®) [41]. The HAV serological status should be assessed before immunization as well as for HBV [5].

Studies in the present literature confirm that HAV vaccination, in patients with IBD, has an immunogenicity similar to healthy controls. Urganci N et al. [36] and Radzikowski A et al. [43] found in their studies that seroconversion in children with IBD is not significantly different compared to healthy controls. No differences were observed in paediatric patients in the seroconversion rate among different treatment groups with azathioprine, 6-mercaptopurine or steroids [37]. Park SH et al. [44] verified that adult patients with IBD have a seroconversion rate similar to healthy adults, however, the authors concluded that patients receiving anti-TNFα agents have a lower seroconversion rate than other patients with IBD.

There are no available data regarding the vaccination rate against HAV in the international literature, and in a recent cohort study, Dimas I et al. [45] found that two-thirds of their patients with IBD was vaccinated against HAV.

### 3.6. Human Papillomavirus Virus (HPV) Vaccination

Vaccination against HPV is recommended to prevent HPV infections and HPV-associated diseases, including cancers. The HPV infection may cause cervical, vaginal, and vulvar cancers in women, penile cancers in men and oropharyngeal, anal cancers and genital warts in both men and women [46]. According t**o** Trottier H et al. [47], 6.2 million persons in the world refer a new infection of HPV annually [47]. An impaired immunological status and smoking attitude increase the risk of dysplasia and cytological progression [48].

The international guidelines for IBD recommend to vaccine both female and male patients from 11 to 26 years as healthy persons [5,6]. The HPV vaccine is inactivated, safe and can be administrated even during immunosuppression therapies [5]. Three formulations are available: a two-vaccine (2vHPV), a four-valent vaccine (4vHPV) and nine-valanet (9vHPV). The recommended immunization schedule is two doses of the HPV vaccine at 0 and 6–12 months before 15 years, and three doses of the HPV vaccine at 0, 1–2 and 6 months after 15 years. The most recent vaccination is the 9vHPV formulation. Vaccination for adults older than 26 years should be evaluated by the physician according to the patient's risk factors for HPV infection [46].

Jacobson DL et al. [49] investigated the effectiveness of the 4vHPV in patients with IBD undergoing anti-TNF$\alpha$ or immunomodulator therapy. Their study successfully demonstrated that the HPV vaccine was immunogenic and no vaccine-associated adverse effects were observed [49].

A meta-analysis evaluated the risk of cervical high-grade dysplasia (HGD) and cervical cancer in 77.116 patients with IBD from five cohort studies and three case–control studies. The authors found an increased risk for cervical HGD or cervical cancer among immunosuppressed patients with IBD compared to healthy controls (OR 1.34; 95% CI 1.23–1,46) [50]. Moreover, the American College of Obstetricians and Gynecologists recommends annual screening for women with a history of immunosuppressive therapy [51]. However, a current or past HPV infection is not a contraindication for immunomodulator therapy [6]. Anal cancer may be a complication in Crohn's Disease and the risk of developing anogenital warts is higher in immunosuppressed patients; in this case drug withdrawals should be discussed in multi-disciplinary teams [6,52].

Waszczuk E et al. [53] reported that only 69% of their female patients with IBD performed a regular PAP-test (30% annually, 32% every 2–3 years, 7% every 5 years) and only 10% of women claimed that the HPV vaccine was recommended for patients with IBD.

### 3.7. Diphtheria, Tetanus and Pertussis Vaccinations

Diphtheria (*Corynebacterium Diphtheria*), Tetanus (*Clostridium Tetani*) and Pertussis (*Bordetella Pertussis*) are three different important diseases which can all be prevented by vaccination. This vaccination can be safely administrated also in patients with IBD, because it is an inactivated vaccine [54]. The ACG guidelines [5] recommend following the ACIP vaccination schedule. Adolescent (11–18 years old) vaccinated during childhood should receive one dose of a tetanus and diphtheria toxoids and acellular pertussis (Tdap) and a booster dose of tetanus and diphtheria toxoids (Td) every 10 years. Adolescents and adults not already vaccinated should receive one dose of Tdap as soon as possible and a booster dose of Td every 10 years [54].

IBD and immunosuppressant drugs can interfere with the immunization and both can lower the antibody concentrations of patients with IBD compared to healthy controls. Caldera et al. [55] reported that pertussis antibody concentrations were lower in patients with IBD than in the controls, and those in treatment with anti-TNF$\alpha$ had a lower antibody concentration than those on thiopurine monotherapy. Moreover, diphtheria antibody concentrations were lower in patients with IBD than in controls, instead no significant difference in tetanus antibody concentrations were observed between IBD group and controls [55]. A prospective controlled trail highlighted that antibody responses to tetanus and pertussis vaccination may be affected by a therapeutic drug regimen and the response rate to vaccination is lower in patients on combined therapy than patients with IBD on monotherapy [56].

Different strategies were proposed to increase both the vaccination rate and seroconversion rate among patients with IBD: to schedule the Tdap vaccination at the same time as the diagnosis or to perform a more aggressive vaccination schedule may improve antibody concentration [55,56].

*3.8. Poliomyelitis Vaccination*

Poliomyelitis is a highly contagious infectious disease caused by poliovirus. Most of poliovirus infections are asymptomatic, but symptomatic cases are typically characterized by fever, while only a small percentage of cases will develop aseptic meningitis or paralytic disease [57]. The ECCO guidelines recommend checking the patient's immunological status for polio vaccination and vaccination should be routinely administered as in the general population [6]. According to the ACIP guidelines, unvaccinated adults should receive three doses of inactivated poliovirus vaccine (IPV) at 0, 1–2 and 7–14 months. IPV is an inactivated vaccine and it can be administrated in patients with immunodeficiency disorders. The oral poliovirus vaccine (OPV) is contraindicated in persons who have immunodeficiency disorders; therefore the vaccination status should be assessed in people with an increased risk of exposure to poliovirus [57].

No data are available regarding immunological protection against Poliomyelitis in patients with IBD after vaccination.

*3.9. Haemophilus Influenzae vaccination*

Haemophilus influenzae type b (Hib) is among the most common pathogens that can cause bacterial meningitis and other invasive diseases in infants, elderly people and immunocompromised patients [58]. Stobaugh DJ et al. [59] demonstrated, in a nationwide cross-sectional study, that patients with IBD have an increased risk compared to controls for hospitalization for pneumonias due to Hib. The ACG guidelines recommend vaccinating patients with IBD against Hib, regardless of the immunosuppression status, because of the Hib vaccine is inactivated and can be safely administered [5]. According to the ACIP guidelines, Hib vaccination is not necessary in adults, unless in splenectomized patients. The suggested schedule for the monovalent vaccine is: two doses at 0 and 8 weeks apart for PRP-OMP (PedvaxHIB®), or three doses at 0, 8 and 16 weeks apart for PRP-T (ActHib®) [58]. Dotan et al. [60] observed that patients with IBD had a normal response to the Hib vaccine even if in therapy with thiopurines. However, no data are available regarding the response of patients with IBD to Hib vaccine and other immunosuppressive therapies.

## 4. Live-Attenuated Vaccines

*4.1. Measles, Mumps and Rubella Vaccination*

Measles, Mumps and Rubella (MMR) are acute viral diseases that can cause serious illness and complications, but they can be prevented by vaccination. The MMR vaccine is live-attenuated and according to the ECCO and AGC guidelines, it should not be administrated during immunosuppressive treatments [5,6]. If the patient is unvaccinated, the vaccination should be administrated at least 6 weeks before starting immunosuppressive therapy according to the ACG guidelines [5], or in accordance with the ECCO guidelines, recommending that at least 3 weeks before is adequate [6]. The vaccination schedule in immunocompetent IBD patients should be the same as one in the general population. The ACIP guidelines suggest that unvaccinated adults should receive at least one dose of MMR vaccine, and people at high risk of mumps infection should receive two doses (1 month apart), whereas all family members and close contacts of an immunocompromised patient should receive two doses of the MMR vaccine (1 month apart), if unvaccinated [61].

Physicians should check the serological level of MMR antibodies at the diagnosis or before starting immunosuppressive therapies, even if their patients were previously vaccinated. In fact, Naganuma et al. [62] discovered that about 30%, 34%, and 37% of their patients had seronegative levels of antibodies for the rubella, measles, and mumps viruses, respectively. Caldera et al. [63] evaluated whether antibody concentrations for MMR were influenced by immunosuppressive

therapy. They discovered that antibody concentrations in the IBD group was similar to those in controls, suggesting that vaccinating patients before treatments was strategically important to keep an adequate immunization level [63].

### 4.2. Varicella Vaccination

Varicella (Chickenpox) is a highly contagious disease caused by the varicella zoster virus (VZV). Varicella primary infection causes a systemic infection that usually results in lifetime immunity, however, VZV remains dormant in sensory-nerve ganglia and may be reactivated during lifetime causing herpes zoster (shingles). Varicella is usually a childhood disease. Primary infection is usually self-limited, but immunocompromised patients have an increased risk of complications [64]. According to the ACG and ECCO guidelines, patients with IBD with unknown status or without previous varicella vaccination should be vaccinated [5,6]. Vaccination should be administered at least 3 weeks prior to immunosuppressive treatments, and must not administered during immunosuppression and can be only administered after 3–6 months from cessation of all immunosuppressive therapy [6]. Testing VZV IgG is also recommended [6], and a recent study found that patients with IBD and a positive history of chickenpox or shingles were seronegative for VZV IgG and exposed to VZV infection or reactivation [65].

The Varicella vaccine is a live-attenuated vaccine and the vaccination schedule for Varicella suggested by the ACIP guidelines in healthy adults is: two doses of single-antigen varicella vaccine administrated 4–8 months apart. Vaccination in immunodeficient patients should be evaluated according to the patient's status and the Varicella vaccine should not administered in patients on 5-amynosalicilate or corticosteroids [64]. All family members and close contacts of an immunocompromised patient should be vaccinated [6,64].

Harris RE et al. [65] evaluated the varicella screening and immunisation programme in children with IBD in the UK, where the vaccination schedule does not include varicella immunization. The authors found that the median age of the diagnosis of naïve patients was lower than that of immunized patients. About 33% of unimmunised patients required post-exposure prophylaxis or treatment for varicella, while prior immunization was associated with a decrease in the need for post-exposure prophylaxis and varicella-related hospital admission [66]. Adams DJ et al. [67] suggested the ideal timing to vaccinate patients with IBD without varicella immunity is before the initiation of immunosuppressive therapy. The authors found that there was an association of IBD and immunocompromising conditions with hospitalization both for varicella and the herpes zoster [67]. However, there is still lack of data on the safety and efficacy of the varicella vaccine in patients with IBD on immunomodulators or biologic therapy.

### 4.3. Herpes Zoster Vaccination

Herpes zoster (HZ) is caused by the reactivation of the VZV in the sensory-nerve ganglia, it is usually painful, and it can develop serious consequences in immunodeficient patients. HZ risk is increased in adults aged ≥60 years and the ACIP guidelines recommend their vaccination [68]. Patients with IBD have a higher risk of developing HZ and HZ-related complications regardless of the duration of the disease. Moreover, HZ is more frequent in young patients with IBD than age-matched healthy controls [69]. Immunosuppressive treatments are an additional risk factor for HZ reactivation in patients with IBD. Khan et al. [70] found that thiopurines, alone or in combination with anti-TNF$\alpha$, increased the risk of HZ, while anti-TNF$\alpha$ drugs alone seem not to be associated with an increased risk.

The AGC and ECCO guidelines recommend vaccination with the HZ live-attenuated vaccine in patients with IBD aged over 50 years, and immunomodulator therapy should not be started during active disease and antiviral or prophylaxis treatment should be considered [5,6]. However, in October 2017, the new ACIP guidelines recommend using recombinant zoster vaccine (RZV) in immunocompetent adults over 50 years, rather than the live-attenuated vaccine. According to the ACIP, RZV can be also administered to persons taking low-dose immunosuppressive therapy, but it is not licensed for those on moderate to high doses of immunosuppressive therapy [68].

A recent prospective observational study studied the risk of adverse reactions and the risk of IBD flare after vaccine administration. The authors observed that the rates of local and systemic adverse reactions were comparable to those seen in the RZV clinical trials and they registered a low rate of IBD flare (1.5%) [71].

Vaccination rates against HZ are low in patients with IBD. Khan et al. [72] found, in a cohort study in the USA, that only 20.96% of patients with IBD were properly vaccinated. These are lower in persons without health insurance. Few data in the scientific literature suggest that RZV is safe and well tolerated in patients with IBD [71]. No data are available suggesting that RZV grants the same protection levels of the live-attenuated vaccine in patients with IBD.

## 5. Cholera, Yellow Fever and Other Vaccinations

According to the ACG and ECCO guidelines, other vaccinations should be considered in patients with IBD who are immunosuppressed and live or are going to travel to areas endemic for a specific infective disease [5,6].

Cholera inactive vaccine (oral inactivated B-subunit-whole-cell) should be administered to patients with IBD, however in colectomized patients, due to ulcerative colitis, the serological responses were lower [73]. Kilhamn et al. [74] suggested that the recombinant cholera toxin B subunit (rCTB) administered into the ileal pouches of patients was capable of inducing a significant IgA antitoxin level in ileostomy fluid. No data are available about the live-attenuated oral cholera vaccine (Vaxchora®) administration to patients with IBD, and data for immunocompromised persons are few [75].

All patients with IBD should be advised to not travel to any areas where yellow fever (YF) is endemic, unless vaccinated. There is no specific treatment for this disease. Steps to prevent yellow fever virus infection also include using insect repellent and wearing protective clothing [76]. In healthy adults, the vaccination schedule is: one dose of live-attenuated vaccine and one booster dose every 10 years. YF vaccine is contraindicated in immunosuppressive and immunomodulatory therapies as well as in immunodeficiency disorders and HIV infection. No specific data exist on the use of YF vaccine in persons receiving these therapies [76].

Estave et al. [77] suggested that patients with IBD should also be considered for other vaccines like those for the Japanese encephalitis (inactivated virus), "Prick–Zecca encephalitis, malaria, tuberculosis and "Bug's Prick" disease and others. The physician should advice the patients, who are going to travel to areas at risk of disease infection, to consult a specialist in tropical and infective disease.

## 6. Vaccination Schedule in New-Borns from Women on Anti-TNFα Therapy

According to the ECCO guidelines, new-borns must receive particular attention in the first months of life to perform a correct immunization. Vaccination with live-attenuated vaccines should be postponed after 6 months of life [6]. In fact, Mahadevan et al. [8] reported that anti-TNFα drugs (such as infliximab and adalimumab) can cross the placenta and can be detected in infants at birth and up to 6 months after birth. Recent evidences suggest that in utero exposure to anti-TNFα drugs does not seem to increase the short-term or long-term risk of severe infections in children [78] or allergy [79] when compared with unexposed children of non-IBD women. Duricova et al. [80] also reported that children exposed to anti-TNFα drugs in utero had a lower response to Hib and MMR vaccination compared to unexposed children. Luu et al. [81], in a nationwide cohort study, did not observe any severe adverse events in new-borns exposed to anti-TNFα agents, but the authors also observed that vaccinations are often not postponed in keeping with the recommendations [81].

Paediatricians should avoid administrating live-attenuated vaccines in the first 6 months of lifetime and postpone their administration [6]. Live-attenuated vaccines scheduled after the first 6 months of lifetime can be administrated as scheduled in healthy children. Inactivated vaccines can be safely administrated, but evidence suggests that new-borns exposed to anti-TNFα agents may have a lower response due to anti-TNFα drug levels after birth [79–81].

## 7. Actual Situations and Future Perspectives

Fatefully, vaccination rates among patients with IBD are lower than those in the general population, despite them have a higher and measurable risk of infective disease. Yeung et al. [82] discovered that only 14.3% of interviewed gastroenterologists were aware of the immunization history of their patients. Moreover, 23.1% of the interviewed gastroenterologists did not know whether live-attenuated vaccines should be avoided by those in the immunosuppressed state. Gurvits et al. [83] interviewed the primary care physicians (PCP) and they found that only 49% of physicians took vaccination history frequently or always and 76% of them reported that they never or rarely checked the immunization antibody levels in patients with IBD. Only 2.5% of the physicians correctly recommended vaccinations all the time (compared to ECCO guidelines) and up to 23% of the physicians would incorrectly recommend live-attenuated vaccines to immunocompromised patients with IBD [83].

Nowadays, different surveys were performed to evaluate the level of knowledge and the opinions of physicians who have the role of administrating vaccination in patients with IBD. Malacuso et al. [84], in a European-based survey, reported a response rate of 43.5% from interviewed physicians. Their results showed a high awareness of the problem; in fact, 82.9% of responders agreed that performing the vaccinations recommended by the guidelines in patients with IBD is "very important", however, only 55.6% of responders performed immunization at diagnosis of IBD [84]. For this reason, the authors argue that there is still a resistance to vaccination, even if recommended by the guidelines [84]. Moreover, Al-Omar et al. [85], in a Saudi-based survey, found similar results. Both working groups came to the conclusion that it is necessary to increase the educational activities promoted by scientific societies and to implement policies and procedures to promote vaccination in patients with IBD [84,85].

Furthermore, patients with IBD are not aware of their risk of infective disease, according to Yeung et al. [82], as 21.7% of them refused to be vaccinated. Nevertheless, patient education strategies have proven usefully; Jackson BD et al. [19] could improve vaccination rate in their patients from 21% to 32% for pneumococcal vaccination and from 23% to 47% for influenza vaccination.

Vaccination schedule in patients with IBD can be and should be planned by the gastroenterologist and by the PCP together with the patient themselves. The gastroenterologist should check their patient's vaccination status at the moment of diagnosis and plan immunosuppressive drugs. They can also prescribe vaccination according to the patient's status. The PCP can vaccinate the patients when the disease is quiescent and their family members/close contracts if is necessary [2–6]. Waszczuk et al. [86] suggested "a cocoon strategy" of vaccination, because the vaccination of close contacts permits to protect vulnerable patients from infectious diseases.

To achieve this goal, it is important that both physicians and patients are aware that vaccinations do not cause any disease; it is a safe medication and does not worsen the severity of the IBD. Vaccination timing is also pivotal, and vaccination should start before immunosuppressive therapy and aging, because both increase the risk of infective diseases [2–6]. Moreover, immunosuppressed patients with IBD aged over 60 years seem to respond less to vaccinations [6].

Another important issue is the differences which exist among the vaccination schedules of each country. In fact, the vaccination schedule of healthy people is defined by the country in which they live and most of these differences mainly concern the paediatric population [87]. This fact may lead to a different access to vaccination. However, patients with IBD should not have different vaccination coverage depending on the country in which they live, as is for healthy people [5,6]. Indeed, the current ACG and ECCO guidelines provide the possibility of vaccinating patients after the diagnosis of IBD, if not vaccinated during childhood or prior the diagnosis [5,6]. Nevertheless, these guidelines do not recommend the pharmacological composition and commercial formula of the vaccine itself. This is because not all formulations are available in all countries. Moreover, not all vaccines recommended by the ACG and ECCO guidelines are reimbursed by the health service or private insurance. This could generate a vaccination hesitation due to the cost of the vaccine. For these reasons, physicians should evaluate for each patients the best suitable vaccine formula available in

their country and they should convince the patient of the cost effectiveness of the vaccine, preferring the formulations recommended by their national vaccination guidelines as suggested by the ACG and ECCO guidelines [5,6].

## 8. Conclusions

Vaccines are one of the most powerful tools we have available to contain and prevent infectious diseases [1]. The risk of infectious diseases, to which patients with IBD are exposed, could be considerably reduced if the vaccination coverage were similar to that in the general population [2–4]. Low vaccination rates and lack of awareness are both two great challenges, which must be wined to achieve this goal. The education of both patients and doctors could be useful to improve vaccination awareness. Moreover, gastroenterologists and PCPs should work together to improve the vaccinal status of patients with IBD. Inactive vaccines can be administered even during immunosuppressive treatments, while the gastroenterologist should evaluate when the best timing is for live-attenuated vaccines. Despite the concerns for impaired immune responses in immunocompromised patients with IBD, most of these patients develop an adequate response after vaccination (Table 2 provides a quick reference guide for vaccinating patients with IBD).

However, this may not be enough, because in the future, ad hoc vaccination schemes may be needed to ensure a constant seroprotection level, as some research teams proposed for HBV vaccination in IBD [36,38]. Finally, new vaccines and a better comprehension of the immunological mechanism behind IBD and immunosuppressive drugs will improve vaccinal strategies in the IBD population.

**Table 2.** Vaccination in inflammatory bowel disease (quick reference).

| Type of Vaccine | Vaccine/Microorganism | Vaccination Schedule |
|---|---|---|
| Inactivated/ Intramuscular | Pneumonia vaccine/ *Streptococcus pneumoniae* | One PVC13, one PPVS23 at 8 weeks and one PPVS23 after 5 years. One booster dose at 65 years. |
| Inactivated/ Intramuscular | Meningitis vaccine/ *Neisseria Meningitidis* | Men-ACWY: one or two doses (2 months apart) Men-B: three doses (at 0, 1–2 and 6 months) One booster dose after every 5 years in high-risk population |
| Inactivated/ Intramuscular | Influenza vaccination/ *Flu viruses A and B* | One dose every year of inactivated IIV3 |
| Inactivated/ Intramuscular | HVB vaccination/ *Hepatitis B virus* | Three doses (at 0, 1 and 6 months) |
| Inactivated/ Intramuscular | HVA vaccination/ *Hepatitis A virus* | Two doses * (at 0 and 6–12 months) Three doses * (at 0, 1 and 6 months) |
| Inactivated/ Intramuscular | HPV vaccination/ *Human papilloma virus* | Before 15 years: two doses (at 0 and 6–12 months) After 15 years: three doses (at 0, 1–2 and 6 months) |
| Inactivated/ Intramuscular | Tdap/Td vaccination/ *Corynebacterium Diphtheria, Clostridium Tetani and Bordetella Pertussis* | Previously vaccinated (11–18 aged): one dose of Tdap vaccine Not prior vaccinated: one dose of Tdap vaccine One booster of Td every 10 years |
| Inactivated/ Intramuscular | Poliomyelitis vaccination/ *Poliovirus* | Three doses of IPV (at 0, 1-2 and 7-14 months) |

| Inactivated/ Intramuscular | Hib vaccination/ *Haemophilus influenzae type b* | Two doses * (at 0 and 8 weeks) Three doses * (at 0, 8 and 16 weeks) |
|---|---|---|
| Inactivated/ Oral | Cholera vaccination/ *Vibrio Cholerae* (Only for endemic areas or travellers) | One dose of oral inactivated B-subunit-whole-cell |
| Live-attenuated/ Intramuscular | MMR vaccination/ *Measles, mumps and rubella viruses* | One or two doses (1 months apart) according to immunological status and risk factors |
| **Live-attenuated/** **Intramuscular** | Varicella vaccination/ *Herpes zoster virus* | Two doses (4–8 months apart) |
| **Live-attenuated/** **Intramuscular** | Herpes zoster vaccination/ *Herpes zoster Virus* | One dose of live-attenuated vaccine or one dose recombinant vaccine |
| **Live-attenuated/** **Intramuscular** | Yellow fever vaccination/ *Yellow fever Virus* | One dose One booster every 10 years |

**Author Contributions:** Concept: G.C., C.M., F.B., A.L.; Investigation: G.C., F.I.; Software: F.I.; Formal Analysis: F.I., A.I. Validation: V.V., F.I.; Methodology: F.I., A.I. Data Curation: F.I., G.C., V.V. Writing on draft preparation: G.C., V.V. Writing Review: F.I., V.V.; Visualization: F.D.S. Project Administration: A.I., G.C.; Supervision: V.V. All authors have read and agreed to the published version of the manuscript.

**Funding:** This research received no external funding.

**Conflicts of Interest:** The authors declare no conflict of interest.

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
