# Peer review of "Vaccination in Inflammatory Bowel Disease: Utility and Future Perspective"

_gastrointestdisord, doi:10.3390/gidisord2020019_

Round 1

Reviewer 1 Report

In the present review article Casella et al discussed about the problem of vaccination in IBD patients, analyzing benefits and risks and proposing possible strategies in cases of immuno-suppressed patients.

Overall, this is a very good paper which covers almost all the aspects of the topic. My only observation is that in children born from IBD mothers under anti-TNF treatment, the vaccination schedule should be tailored even if the newborn is not affected by IBD, because anti-TNF drugs can pass through the placenta (please see guidelines). Therefore a dedicated paragraph is necessary.

A short paragraph describing the level of knowledge and the opinions of physicians may improve the paper (see Macaluso FS et al, Digestion 2019).

There are some typos which should be amended, for example page 1 line 39: IDB - -> IBD.

Author Response

Dear Reviwer

Thank you very much for your comments to the manuscript, entitled: “Vaccination in Inflammatory Bowel Disease: Utility and Future Perspective”. Here you will find in detail the changes that the authors made to the manuscript after the reviewers' comments:

  1. Line 38: substitution of “IDB” with “IBD”;
  2. Line 420 to 437: insertion of a new paragraph, dedicated to vaccination schedule in new-borns from women on anti-TNFα therapy, the new paragraph numbering is 6;
  3. Insertion of four new bibliographical references for paragraph 6 (81, 82, 83, 84);
  4. Line 438: paragraph numbering changed from 6 to 7 (due to new paragraph 6);
  5. Line 443: reference numbering changed from 81 to 85 (due to new references);
  6. Line 448: reference numbering changed from 82 to 86 (due to new references);
  7. Line 449 to 458: insertion of a new paragraph with the aim of briefly describing the level of knowledge and awareness of physicians on the topic;
  8. Insertion of two four new bibliographical references for paragraph (line 449 to 458) (87, 88)
  9. Line 460: reference numbering changed from 81 to 85 (due to new references);
  10. Line 496: reference numbering changed from 83 to 89 (due to new references);
  11. Line 475: paragraph numbering changed from 7 to 8 (due to new paragraph 6);

Looking forward to hearing from you in due course,

On behalf of all co-authors

Yours sincerely,

Giovanni Casella, MD

Reviewer 2 Report

During IBD a majority of patients requires long-term immnosuppressive therapy with immunomodulatory agents, biologics or immunosuppressive therapies (Vedolizumab, UStekinumab..). Due to this, they are at increased risk for infectious diseases, many of which are possible to prevent by vaccination. Therefore, this review summarizes the vaccination recommendations for IBD patients and provides a useful guide in managing vaccination schedule. 

The review is very interesting, the vaccination status should be assessed as soon as the disease is diagnosed and  vaccinations should be administered, ideally before immunosuppressive therapy is started.

The manuscript is nicely written, the structure is appropriate and references are adequate. 

Author Response

Dear Reviwer,

Thank you very much for your comments to the manuscript, entitled: “Vaccination in Inflammatory Bowel Disease: Utility and Future Perspective”. Here you will find in detail the changes that the authors made to the manuscript after the reviewers' comments:

  1. Line 38: substitution of “IDB” with “IBD”;
  2. Line 420 to 437: insertion of a new paragraph, dedicated to vaccination schedule in new-borns from women on anti-TNFα therapy, the new paragraph numbering is 6;
  3. Insertion of four new bibliographical references for paragraph 6 (81, 82, 83, 84);
  4. Line 438: paragraph numbering changed from 6 to 7 (due to new paragraph 6);
  5. Line 443: reference numbering changed from 81 to 85 (due to new references);
  6. Line 448: reference numbering changed from 82 to 86 (due to new references);
  7. Line 449 to 458: insertion of a new paragraph with the aim of briefly describing the level of knowledge and awareness of physicians on the topic;
  8. Insertion of two four new bibliographical references for paragraph (line 449 to 458) (87, 88)
  9. Line 460: reference numbering changed from 81 to 85 (due to new references);
  10. Line 496: reference numbering changed from 83 to 89 (due to new references);
  11. Line 475: paragraph numbering changed from 7 to 8 (due to new paragraph 6);

Looking forward to hearing from you in due course,

On behalf of all co-authors

Yours sincerely,

Giovanni Casella, MD